# The Simultaneous Detection of Multiple Antibiotics in Milk and Pork Based on an Antibody Chip Biosensor

**DOI:** 10.3390/bios12080578

**Published:** 2022-07-29

**Authors:** Jiaxu Xiao, Nana Wei, Shuangmin Wu, Huaming Li, Xiaoyang Yin, Yu Si, Long Li, Dapeng Peng

**Affiliations:** 1National Reference Laboratory of Veterinary Drug Residues (HZAU) and MOA Key Laboratory for the Detection of Veterinary Drug Residues in Foods, Huazhong Agricultural University, Wuhan 430070, China; xiao_jiaxu@163.com (J.X.); wsm3510346@163.com (S.W.); lihm9915@163.com (H.L.); yinxy73251@163.com (X.Y.); siyu07075529@163.com (Y.S.); 15207165483@163.com (L.L.); 2Kunpeng Institute of Modern Agriculture at Foshan, Foshan 528200, China; weinana1910@163.com; 3Shenzhen Institute of Nutrition and Health, Huazhong Agricultural University, Shenzhen 518000, China; 4Shenzhen Branch, Guangdong Laboratory for Lingnan Modern Agriculture, Agricultural Genomics Institute at Shenzhen, Chinese Academy of Agricultural Sciences, Shenzhen 518000, China; 5Genome Analysis Laboratory of the Ministry of Agriculture, Agricultural Genomics Institute at Shenzhen, Chinese Academy of Agricultural Sciences, Shenzhen 518000, China

**Keywords:** multi-residues, antibody chip, cephalosporins, aminoglycosides, sulfonamides

## Abstract

In the modern farming industry, the irrational or illegal use of veterinary drugs leads to residues in animal-derived food, which can seriously threaten human health. Efficient detection of low concentrations of drug residues in animal products in a short time is a key challenge for analytical methods. This study proposes to use an antibody chip biosensor for rapid and automated analysis of cephalosporins, aminoglycosides, and sulfonamide antibiotics in pork and milk. 3D polymer slides were applied for the preparation of antibody chips. Ovalbumin (OVA) or bovine serum albumin (BSA) conjugates of the haptens were immobilized as spots on disposable chips. Monoclonal antibodies (mAbs) against cefalexin, ceftiofur, gentamicin, neomycin, and sulfonamides allowed the simultaneous detection of the respective analytes. Antibody binding was detected by a second antibody labeled with Cy3-generating fluorescence, which was scanned a with chip scanner. The limits of detection (LOD) for all the analytes were far below the respective maximum residue limits (MRLs) and ranged from 0.51 to 4.3 µg/kg. The average recoveries of all the analytes in each sample were in the range of 81.6–113.6%. The intra- and inter-assay CV was less than 12.9% and showed good accuracy and precision for all the antibiotics at the MRL level. The sample pretreatment method is simple, and the results are confirmed to be accurate by LC–MS/MS; therefore, this method is valuable for the quality control of animal-derived food.

## 1. Introduction

Cephalosporins, aminoglycosides, and sulfonamide antibiotics have been widely used in veterinary clinics and animal husbandry to prevent and treat bacterial infections [1,2]. These antibiotics are seriously abused for economic benefits, resulting in increasingly serious drug residues in animal-derived food and the production of large numbers of drug-resistant bacteria, which directly threaten human health [3,4,5]. Therefore, regulatory agencies around the world have established maximum residue limits (MRLs) for veterinary drugs in animal-derived food [6] (Commission 2009) (Commission Regulation, No 470/2009).

Rapid and efficient detection of drug residues in animal-derived food remains a key issue. Several residue detection methods have been reported, such as chromatographic methods [7,8], microbiological methods [9,10,11], and enzyme-linked immunosorbent assays (ELISA) [12,13,14]. Microbiological methods in residue screening on high throughput showed good performance but lack specificity and sensitivity. Instrument methods with high sensitivity are often used as confirmation methods but require time-consuming sample pretreatment, bulky and expensive instruments, and accurate operation, which severely limit their wide application. The ELISA method is simple, sensitive, and low-cost; but only one class of drugs can be detected at a time, and the detection efficiency is not high. The biosensing technology based on antibody chips perfectly addresses this requirement. By enriching different antigen–antibody reactions on the chip and by collecting immunofluorescence sensing signal values from different chips, it is realized that multiple drugs can be screened simultaneously in a large number of samples [15,16]. This method also retains the advantages of rapidity, sensitivity, and specificity of traditional ELISA analysis, which greatly improves the detection efficiency. In addition, the simplicity of the operating procedures of antibody chip technology makes it more favorable for widespread application and promotion [17]. These important characteristics make antibody chip biosensors an attractive tool for food testing by regulatory agencies to ensure food safety.

Currently, many researchers have developed several antibody chip biosensors which are potentially useful for antibiotic multiplexed analysis. O’Mahony et al. developed a chemiluminescence-based biochip array for the rapid analysis of nitrofuran antibiotics in honey [18]. Wang et al. developed an immunochip for the visual semi-quantitative detection of six fungal toxins in 4 h with minimal use of samples [19]. Zhao et al. presented a disposable MoS2-arrayed matrix-assisted laser desorption/ionization mass spectrometry (MALDI MS) chip for the quantitative detection of multiple sulfonamides, with 96 sample spots analyzed on one single chip [20]. CapitalBio Corporation developed protein chip kits I and II for veterinary drug residue detection. Randox Laboratories Limited developed a biochip array technology systems evidence investigator™, which can determine multiple drug residues in massive samples simultaneously. However, their detection process, sensitivity, and specificity as well as the kinds of drugs tested need to be further improved. And to the best of our knowledge, there are no reports of antibody chips for multi-residue analysis of cephalosporin antibiotics.

This study presents a development of an antibody chip system for high-throughput testing and routine screening of antibiotic residues in animal-derived food. We constructed antibody chips by using artificial antigens (conjugation of carrier proteins to small molecule drugs) as probes immobilized on 3D polymer slides. When drugs were present in the test sample, they competed with artificial antigens immobilized on substrates for specific reactions with anti-drug antibodies. The antigen–antibody complexes are detected with Cy3-labeled secondary antibodies and analyzed for fluorescence signals from each test region. The antibodies that could recognize the same type of antibiotic are mixed together to increase the throughput of the antibody chip. We have also established matched sample pretreatment methods with good recoveries and high reproducibility. As expected, the antibody chip biosensor is suitable for the detection of a large number of veterinary antibiotics that have high sensitivity and specificity for the analyte. 

## 2. Materials and Methods

### 2.1. Chemicals and Reagents

Cefadroxil (CFR), cephradine (CE), desfuroyl ceftiofur (DFC), cefquinome (CQO), Ceftriaxone (CRO), sulfadiazine (SD), sulfamerazine (SM1), sulfamethoxypyridazine (SMP), sulfaquinoxaline (SQ), sulfachlorpyridazine (SPD), sulfachloropyrazine (SPZ), sulfadimethoxine (SDM), sulfamethylthiadiazole (SMTZ), sulfadoxine (SDM′), sulfamethoxydiazine (SMD), sulfathiazole (ST), sulfisomidine (SM2′), neomycin (N), gentamicin (GM), amikacin (AK), paromomycin (PRM), and sisomicin (SC) standards were purchased from Germany (Dr. Ehrenstorfer GmbH). Cefotaxime (CTX), sulfamethoxazole (SMZ), sulfamonomethoxine (SMM), and sulfafurazole (SIZ) standards were purchased from Sigma-Aldrich (USA). Sulfadimidine (SM2) and cephalexin (CLX) standards were purchased from TCI. Ceftiofur (EFT) standards were purchased from Sigma.

Six drug conjugates (Cefalexin-OVA, Ceftiofur-OVA, Neomycin-OVA, Gentamicin-OVA, and sulfonamides-OVA/BSA) and their monoclonal antibodies (mAbs) were prepared in the National Reference Laboratory of Veterinary Drug Residues (HZAU). Cy3-labeled goat anti-mouse IgG was purchased from GE (Fairfield, CT, USA). The Cy3-labeled BSA were bought from Biosynthesis Bioss (Beijing, China). The 3D polymer slides were purchased from CapitalBio Corp (Beijing, China). The agarose was purchased from GENE COMPANY LID. The poly-lysine slides were bought from Sangerbio (Shanghai, China). The Superfrost Plus slides were purchased from Fisher (Waltham, MA, USA). The aldehyde slides were purchased from Yadabio (Wuhan, China). All of the other chemicals used were of analytical grade and were provided from the Sinopharm Chemical Reagent Co., Ltd. (Beijing, China).

### 2.2. Apparatus

The BioJet Plus™ AD3200 spotter was purchased from Biodot (Irvine, CA, USA). The InnoScan 700A microarray Laser Scanner was purchased from Innopsys (Chicago, IL, America).

### 2.3. Preparation of Antibody Chip Biosensors

#### 2.3.1. Preparation of Agarose Substrate

First, the slides were sonicated in dilute sulfuric acid and medical alcohol solution for 12 h, then washed with distilled water and dried in an oven at 37 °C for 5 h. Then the slides were activated at 60 °C for 30 min, followed by 2 mL of agarose solution (1%) being poured over each slide. After the agarose had solidified, the slides were dried overnight at 37 °C and then stored at 4 °C for a long time until use. However, when immobilizing the coating antigen, this agarose substrate needed to be reactivated by soaking in 20 mmol/L NaIO_4_ for 30 min, rinsed well with triple distilled water, and dried before use.

#### 2.3.2. Preparation of the Antibody Chips

The conjugates (drug–OVA/BSA) were printed onto the chip substrate (agarose substrate) with BioJet Plus™ AD3200 spotter 500 µm diameter and 1 mm pitch. The concentration of each printed protein was 6–20 µg/mL in the printing buffer. Twelve 3 × 3 arrays were printed in a 2 × 6 grid pattern on each slide. OVA/BSA (1 mg/mL) and Cy3-BSA (5 µg/mL) were used as a negative control and a protein control, respectively, and three replicates were made for each control spot. The upper six arrayed controls were used to prepare the calibration curves, while the lower six arrayed conjugates were used for the determination of the samples. (It is also possible to use the gasket system (CapitalBio) to separate each slide into six separate reactions so that up to six different tissue samples can be analyzed in parallel. It is printed in a cabinet at 25 °C and 60% humidity and the coating antigen is fixed by a Schiff’s base reaction at 37 °C for 0.5 h in a humidified chamber.) Then the antibody chips were incubated in BSA (2% *w*/*v*) in phosphate-buffered saline (PBS, 0.01 mol/L, pH 7.4) for 0.5 h to block the other reactive sites of the substrate. Finally, the chips were rinsed in a phosphate Tween buffer solution (PBST, 0.1 mol/L PBS containing 0.1% Tween-20) and stored dry at 4 °C until use.

### 2.4. Establishment of Calibration Curve for Antibody Chip Biosensor

The process of this biosensor assay is similar to the ic-ELISA analysis but takes less time. First, the antibiotic standard solution was diluted with PBS to a concentration of five suitable gradients. A mixture of 10 µL antibody and 10 µL standard solution was then applied to the grid reaction chamber, covered with antibody chips, and incubated for 0.5 h at 37 °C in a humid chamber. Then the chip was rinsed three times with PBST, and 20 µL of Cy3-labeled goat anti-mouse IgG was added. After another 30 min of incubation and a clean wash, the signal could be scanned. The fluorescent signals (Flu) were collected by the InnoScan 700A Laser Confocal Scanner and initially analyzed with Mapix software (Innopsys). The data were fitted using the common logarithm (lg) of the antibiotic concentration as the X-axis and the corresponding Flu as the Y-axis. The calibration curves of the different antibiotics were generated with GraphPad Prism 6.0 software (GraphPad Software, La Jolla, CA, USA). For each chip, a calibration curve was generated in parallel with the sample analysis and subsequently used for the quantification of each drug residue in the samples. The calibration curve is linear over a given range, and the drug residues corresponding to the Flu of each sample can then be read from the calibration curve.

### 2.5. Validation of Antibody Chip Biosensors

The milk and pork were purchased from a local market in Wuhan and were authenticated to be free of these antibiotics by LC–MS/MS. The residues of sulfonamide and cephalosporin antibiotics in milk can be extracted directly with methanol (10% *v*/*v*) and PBS (pH 8.0), respectively. The 2 mL milk sample was vortexed for 2 min, diluted 30 times with the extraction solution, and centrifuged at 12,000× *g* for 10 min; then the upper fat particles were removed for direct detection. The aminoglycoside antibiotics in the milk needed to be diluted 6 times with trichloroacetic acid (3% *v*/*v*) and centrifuged at 12,000× *g* for 5 min; then the intermediate liquid was diluted 10 times again with PBS (pH 8.0) for detection. The cephalosporins, sulfonamides, and aminoglycoside antibiotics in the pork were extracted with methanol (10% *v*/*v*), PBS (pH 8.0), and trichloroacetic acid (3% *v*/*v*), respectively. Generally, 2.00 g of the homogenized pork sample was added with 10 mL of extract solution and vortexed for 10 min, centrifuged at 6000× *g* for 5 min, and diluted 10 times with PBS (pH 8.0) for detection.

Twenty blank samples were measured, and their concentrations (C) and standard deviations (SD) were calculated separately according to the calibration curve. LOD and limit of quantitation (LOQ) were calculated according to the equations LOD = C + 3 × SD and LOQ = C + 10 × SD. The antibody chip biosensors were validated in pork and milk, spiked at three levels of 1/2 × MRL, 1 × MRL, and 2 × MRL (concentrations of 50, 100, and 200 μg/L in milk and 100, 200, and 400 μg/kg in pork) for each drug. Then, the intra-assay coefficient of variation (CV) was tested on three replicates at each concentration, and the inter-assay CV was measured on three consecutive days to evaluate the accuracy and precision of the biosensors.

To determine the reliability of the developed antibody chip biosensor, a comparison of the samples was performed using both biosensor and LC–MS/MS on four spiked milk and pork samples. In this study, cephalosporins were selected as spiked drugs for validation analysis. LC-MS/MS analysis and the detailed procedure of sample preparation were performed according to the description of Li et al. [21]. According to the analysis of the spiked samples, linear regression curves were plotted to evaluate the correlation between the biosensor and the LC–MS/MS.

## 3. Results

### 3.1. Optimization of the Antibody Chip Biosensor Detection System

#### 3.1.1. Optimization of Agarose-Modified Slides

It was found that the concentration and thickness of the agarose directly affected the homogeneity of the sample spots and the intensity of the relative fluorescence signal, which in turn led to dramatic changes in the performance of the biosensor. Therefore, the agarose concentration was optimized, and Figure 1A demonstrates the change in relative Flu of the same slide with different concentrations (0.4%, 0.6%, 0.8%, 1%, and 1.2%) of agarose modifications. It is clearly observed that the Flu increases with the increase in agarose concentration, especially between 0.4% and 1%. Moreover, the Flu relative to 1% and 1.2% agarose-modified slides was about the same and basically reached saturation, so 1% agarose was chosen to prepare the substrates.

Next, the effect of different thicknesses of agarose (expressed as the amount of spread, 0.5, 1, 1.5, 2, and 2.5 mL) on the adsorption and homogeneity of the samples was investigated (Figure 1B). It can be seen that the signal intensity increases with the increase in the spreading volume but decreases significantly at 2.5 mL. Therefore, the 2 mL with the highest signal intensity was chosen as the spreading amount, and it was assumed that the fluorescence signal was not collected completely because it was too thick. This is the first time that different thicknesses of agarose have been found to have a significant effect on sample adsorption in the antibody microarray preparation procedure. This may be caused by the fact that an overly thick agarose coating blocks the activation of hydroxyl groups on its surface.

#### 3.1.2. Optimization of Antibody-Chip Substrates

Antibodies, proteins, or other antigens are highly complex molecules, so it is necessary to immobilize them in a way that results in efficient deposition and good biological activity [22]. Due to the different structural properties of the substrates, the adsorption capacity of the coated antigens is different, and the antigen–antibody reactions that occur on the biochips are also different. Suitable substrates are crucial for preparing antibody chips. Glass is well suited as a carrier for the preparation of antibody chips due to its low cost and stable surface properties. Two-dimensional (2-D) substrates such as aldehyde slides and poly-lysine slides are commonly used. Three-dimensional (3-D) substrates such as agarose slides and polymer slides G can form a porous mesh structure to absorb more protein. Superfrost Plus slides can adsorb protein firmly by electrostatic effect. 

In this study, these five substrates were used to optimize to find the most suitable substrate. Figure 2 and Figure 3A indicate that the Flu of polymer slides G was higher than other substrates. The background value of polymer slides G was lower than that of aldehyde slides but higher than other substrates (Figure 2 and Figure 3B). The previously reported antibody microarray substrates all use modified agarose substrates. But despite optimization of agarose-modified substrates for maximum strength, only the surfaces of commercialized polymer slides G fulfilled the required immunochip criteria of strong signals, low variance, and background value [23,24,25]. The polymer slide G showed the best adsorption to antigens, followed by the Superfrost Plus slides, then the agarose slides, and finally the aldehyde slides and the poly-lysine slides. 3-D substrates compared to 2-D substrates had shown efficient immobilization when used in conjunction with Flu detection. This conclusion was in accordance with previous research [23,24]. 

Theoretically, it is easy to see this conclusion: first, 3-D substrates can form reticular formation to increase the sample size; second, the gel structure can provide an aqueous environment to keep the biological activity of the protein molecules; third, the aqueous environment is more conducive to antigen-antibody reactions [25]. The adhesivity of the Superfrost Plus slides and the poly-lysine slides is not too good, but the background value is very low, so they are appropriate for fabricating protein chips with a lower background. Certainly, the same substrate has different adhesion properties for different probes, so it is important to find the best substrate.

#### 3.1.3. Optimization of Biosensor Spotting Sequence

It is well known that different spotters have different spotting sequence settings. The different spotting sequences lead to different vibrations of the spotting platform, and the stability of the antigen signal adsorbed to the substrate will be affected by the uniformity of the printed sample dots. Moreover, different protein chips have a different fitting to the spotter, which will achieve different results. Unconformity of the spots on the chip can slow down the matching speed and prolong the data analysis time. If some spots are extremely irregular, it will cause confusion in the data extraction and make the result analysis unreliable. Therefore, four different spotting sequences (Figure 4) were designed for spotting in this study through several experiments, aiming to maximize the performance of the antibody chip biosensor. Figure 4F clearly shows that program C has the highest relative fluorescence signal value and is basically in a stable state with the smallest coefficient of variation, so program C is used as the chip spotting order.

#### 3.1.4. Optimization of Printing and Immobilization

It was reported that the spotting buffer had a significant effect on the adsorption capacity, stability of printed proteins, and spot morphology of the polymer slides G [26]. First, we tested six spotting buffers including PBS, Tris-HCl buffered saline (TBS), carbonate buffer solution (CBS), PBSG (PBS + 10% glycerol), TBSG (TBS + 10% glycerol), and CBSG (CBS + 10% glycerol) (Figure 5A). The sulfonamides-OVA was separately diluted by the six kinds of buffer described above in 10 µg/mL. Meanwhile, we also printed BSA as the negative control and Cy3-OVA as the positive control. The results showed that the CBSG buffer as the spotting buffer produced the highest signal and the best morphology. Second, the pH of CBS was optimized from 9.16 to 10.83 (Figure 5B). With an increase in the pH of CBS, the Flu decreased generally. Third, six surfactants were added to the CBS. Figure 5C showed that the Flu of CBS (pH 9.16) with 0.5% trehalose, CBS (pH 9.16) with 1.5% mannitol, and CBS (pH 9.16) were nearly the same and higher than other spotting buffers. Considering the low concentration of glycerol (5–10%) to reduce the volatilization of the sample spots, therefore, 10% glycerol and CBS (pH 9.16) were chosen as the spotting solution in all the subsequent experiments in this study. In the CBS conditions, the polymer on the slides might form more mesh structures to capture more proteins. The relevant literature reported that 3-D substrates had the best adsorption capacity in a neutral environment and that different probes might cause the contradiction between them [27].

#### 3.1.5. Optimization of Reagent Concentrations for an Indirect Competitive Assay

Based on the immunological analysis, different antibodies have different abilities to bind to antigenic epitopes. The sensitivity of the biosensor is highest only when the coating antigen, mAb, and antibiotics are in a state of balanced competition [28]. The fluorescent signal for the drugs decreased when the drugs were present in the samples. If the mAb is in excess, competition is meaningless, while if there is too little mAb, the antibody–antigen reaction will be incomplete. As a result, the concentration of mAb is an important factor in the quantitative analysis of indirect competition assays. The optimal dilution of antibodies and antigen was determined by a titration test. The Flu between signals of the spot without antibiotics for different antibodies (different dilutions) and half maximal inhibitory concentration (IC_50_) was used for the evaluation. Table 1 shows the optimal concentration combinations of antibodies and antigens for the five antibody chips (detailed data are in Appendix A). We also optimized the dilution of the Cy3-labeled secondary antibody, and the 1:200 dilution was closer to its maximum response. The optimal concentration is different for different antigens and antibodies, and the selection of the most suitable concentration is necessary for the preparation of antibody chip biosensors.

### 3.2. Performance of the Antibody Chip Biosensor

#### 3.2.1. Analysis of Standard Curves and Cross-Reactivity for Biosensors

This study adopted an indirect competitive immunoassay for the quantitative measurement of the antibiotics. The artificial antigens immobilized on the slides bind competitively to the antibodies with the antibiotics (from standard solutions and sample extraction solutions) added to the array. As can be seen in Figure 6, excellent linearity was obtained for each drug, indicating a good fit of the calibration curves. The IC_50_, standard limits of detection (LOD), linear ranges, and MRLs of the five antibiotics under optimal assay conditions are listed in Table 2. The LODs of the five analytes ranged between 0.51 µg/L and 4.30 µg/L, all well below their respective MRLs.

Different antibodies would have different recognition abilities for the same class of target analytes, and the cross-reactivity (CR) of these five different antibody chips for other analytes of the same class is shown in Table 3. Some drugs with similar cross-reactivity mean that the antibody can recognize them simultaneously; for example, the CLX antibody chip biosensor can detect the residues of CLX, CFR, and CE in the samples at the same time. For the detection of aminoglycoside antibiotics in food, the neomycin antibody chip could detect not only neomycin but also amikacin and paromomycin. The gentamicin antibody chip showed low CR with other aminoglycoside antibiotics except for sisomicin (36.3%), indicating that the assay has high specificity. The sulfonamide antibody chip biosensor can simultaneously detect most of the currently reported sulfonamides (16 sulfonamides in total). Thus, the combination of antibody chips could achieve multi-residue monitoring of drugs in animal-derived food.

#### 3.2.2. Sensitivity of Biosensors and Sample Analysis

To determine the effectiveness of the assay, we investigated the LOD, LOQ, accuracy, and precision of the antibody chip biosensor. Based on the calibration curves, Table 4 lists the CVs, LODs, and LOQs of the antibiotics in the samples tested (detailed data are in Appendix A). The average recoveries of all the analytes in each sample were in the range of 81.6–113.6%. The intra- and inter-assay CV were less than 12.9% and showed good reproducibility for all antibiotics at the MRL level. Moreover, both LODs and LOQs were much lower than MRLs, and the antibody chip biosensor was sufficient for trace residue detection.

In addition, the contaminated milk and pork samples were analyzed simultaneously using the biosensor and the LC–MS/MS. Figure 7 demonstrates the fitted correlation coefficients (R^2^) of 0.9917 and 0.9935 for the analytical results of the two methods in the milk and pork samples, respectively. These results demonstrate the high accuracy of the developed antibody chip biosensor and indicate that the biosensor is a reliable tool for the detection of drug residues in animal-derived foods.

#### 3.2.3. Stability of the Antibody Chip Biosensor Assay System

To determine the stability of the antibody chip biosensor assay system, the cephalosporins antibody chip biosensor and reagents were randomly assigned to storage at 37 °C and 4 °C (other biochips in Appendix A). The Flu and sensitivity levels were tested every day for the first group and every month for the second. The hapten–antigen chips and reagents maintained high response values when stored at 37 °C for 7 days or at 4 °C for 6 months (Figure 8). Apparently, the biosensor has good stability without loss of sensitivity and specificity.

## 4. Discussion

In this study, the principle of preparing the antibody chips was the indirect competition method, which does not need to label antigens or antibodies, making the preparation process simpler. However, it requires a two-step reaction of first and second antibodies to detect drug residues in food. The prolonged operation time caused by the two-step reaction was overcome by the combined optimization of the antigen–antibody concentration, the reaction time, and the reaction temperature. At the same time, we have comprehensively optimized the sample pretreatment, making it faster, more efficient, and more environmentally friendly. The antibody chip biosensor reduced the overall detection time from ordinarily 1 day or more to 1.5 h, significantly improving the efficiency of the assay compared to previous reports [29,30].

The type of drug detected by the biosensor depends directly on the performance of the antibody. There are no reports of cephalosporin antibody chip detection methods at home and abroad owing to the lack of corresponding antibodies. This study aims to fill this gap and enrich the kinds of antibody chip biosensors. Research in our laboratory reported an antibody microarray assay for the identification of 6 β-lactams, but a related study reported a receptor-based ELISA for the identification of 15 β-lactams and a metabolite in food [31]. We can possibly immobilize the different receptor proteins on slides to construct reporter chips to broaden the detection of the drugs. Compared to the automated microarray system (IC_50_ = 75 μg/L) developed by Bertram et al. for the simultaneous detection of antibiotics in milk, the IC_50_ of the neomycin antibody chip obtained in this study was 1.97 µg/L, which was nearly 10 times more sensitive for the detection of neomycin [32]. The sensitivity of the gentamicin antibody chip obtained in this study was also greatly improved compared to the parallel affinity sensor array (PASA) developed by Bertram et al. [32]. For the detection of sulfonamide antibiotics, Li et al. printed a protein microarray for the drug residue detection and could only detect sulfamethazine [28]. The antigen microarray system presented by Klaus et al. can recognize sulfadiazine and sulfadimethoxine [33]. As a comparison, the sulfonamide antibody microarray sensor in this study can recognize 13 commonly used sulfonamides simultaneously. 

It is known that the more monoclonal antibodies used, the more types of antibiotics are tested. However, in order to take the detection sensitivity of all drugs into account, the optimization conditions of the entire detection system are more stringent. In previous reports, most use 2–3 monoclonal antibodies to detect fewer than 10 antibiotic residues. However, in our study, the sensor we develop using 5 monoclonal antibodies can detect residues of more than 25 antibiotics simultaneously [17,25,26,27]. The detection range is wider, making it easier to monitor and screen for antibiotic residues.

Compared with traditional rapid drug residue detection techniques, our detection assay has the following advantages: First, each antibody chip can analyze multiple antibiotics and has higher sensitivity and specificity. Second, the high-throughput assay reduces the overall assay time from the usual 1 day or more to nearly 1.5 h. Moreover, the antibody chip biosensor in this study is simple and convenient, with fewer types of reagents and higher accuracy and precision. All these features make this detection system an attractive option for the detection of drug residues compared to previous reports. However, the system requires a scanner to collect the signal from the biochip, resulting in the biosensor not being suitable for on-site testing. Therefore, the trend toward visualization, miniaturization, integration, and automation of biochips is becoming more and more evident. It is necessary to develop a better vehicle to increase the number of test samples at a time. The variety of antibody chips used to detect veterinary drug residues cannot meet the needs of all customers, and the provision of customized services is also a major trend in the future development of the biochip industry.

## 5. Conclusions

In this study, we describe five antibody chip biosensors for the determination of antibiotic residues in milk and pork. An antibody–hapten chip system was implemented using antibodies with high affinity for individual antibiotics, which provides high specificity and very high sensitivity for antibiotic residue analysis. By optimizing the detection system, 3-D polymer slide G shows effective immobilization when used in combination with fluorescence detection, and a spotting buffer CBS containing 10% glycine is a better blocking buffer. Each biochip sensor analysis system is designed to be highly specific, sensitive, and highly accurate and can be easily automated when high throughput is required. Therefore, the antibody chip is very suitable for customs, immigration inspection, quarantine, quality control, and food safety supervision.

## Figures and Tables

**Figure 1 biosensors-12-00578-f001:**
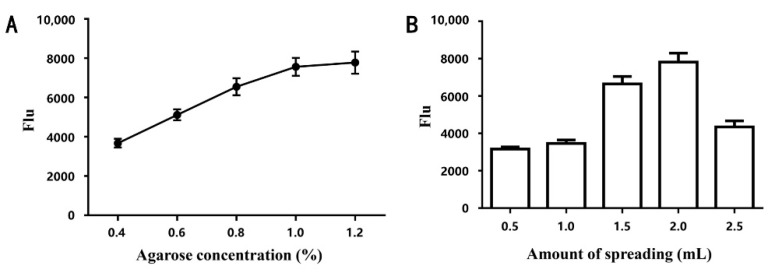
Optimization of agarose concentration (**A**) and amount of spreading (**B**) for agarose-modified slides.

**Figure 2 biosensors-12-00578-f002:**
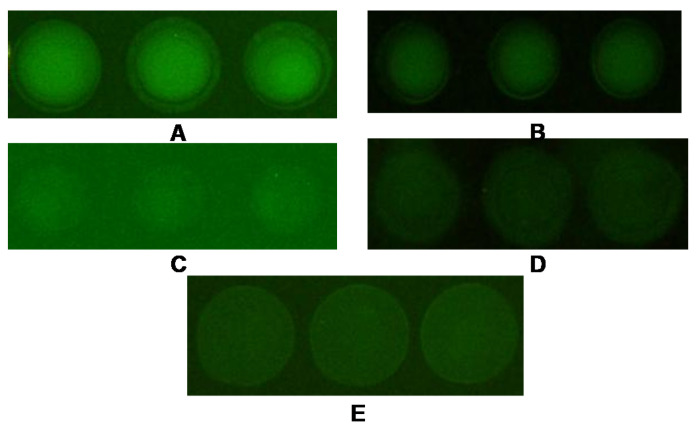
The scanned image of different substrates. (**A**) polymer slide G; (**B**) Superfrost Plus slide; (**C**) aldehyde slide; (**D**) poly-lysine slide; (**E**) agarose surface-modified slide.

**Figure 3 biosensors-12-00578-f003:**
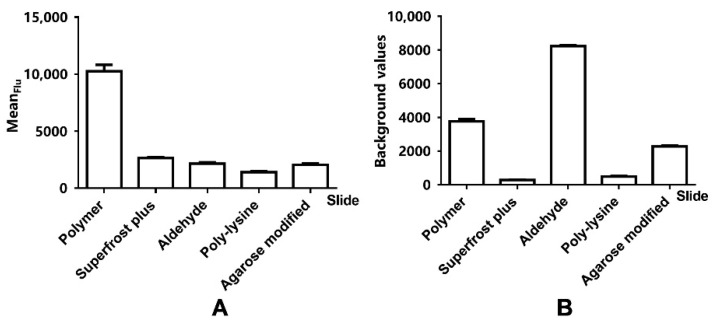
The effect of different substrates on the Flu (**A**) and background (**B**).

**Figure 4 biosensors-12-00578-f004:**
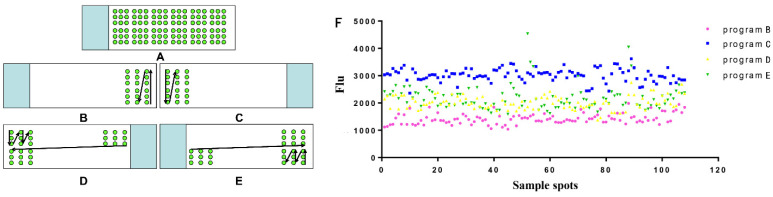
Optimization of biosensor spotting sequence. (**A**) Diagram of the spot sample effect; (**B**–**E**) Four spotting sequence programs; (**F**) The effect of different spotting sequences on sample spots.

**Figure 5 biosensors-12-00578-f005:**
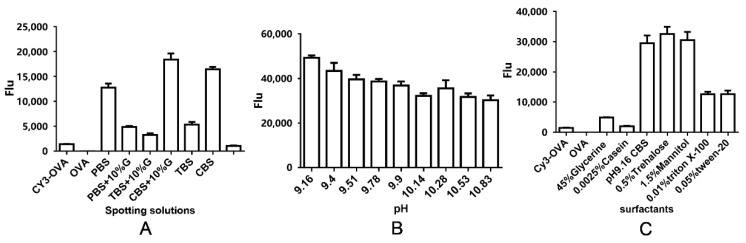
The effect of spotting solutions on Flu.

**Figure 6 biosensors-12-00578-f006:**
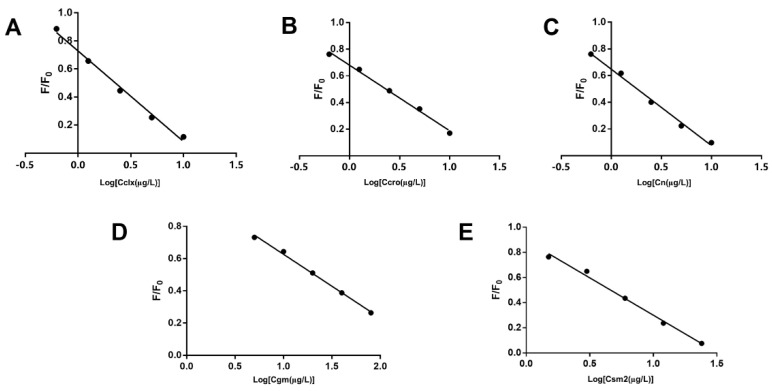
The calibration curves of CLX (**A**), CRO (**B**), N (**C**), GM (**D**), and SM2 (**E**).

**Figure 7 biosensors-12-00578-f007:**
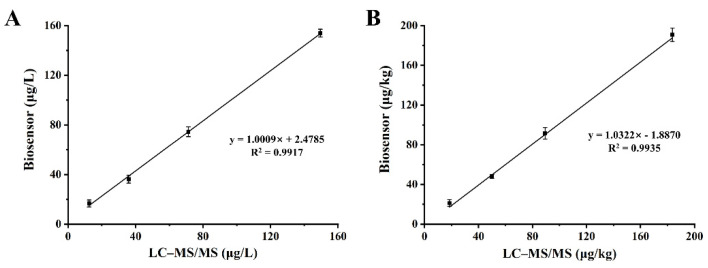
Correlation between the biosensor and the LC–MS/MS for analysis in milk (**A**) and pork (**B**) samples.

**Figure 8 biosensors-12-00578-f008:**
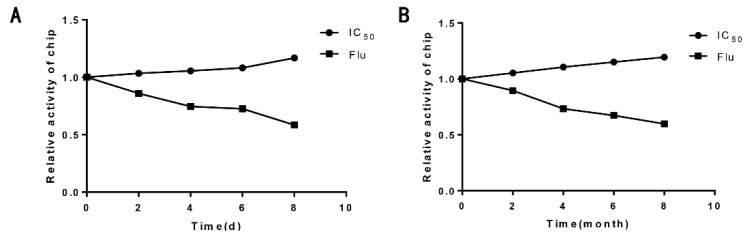
Stability of cephalosporins antibody chip biosensor at 37 °C (**A**) and 4 °C (**B**).

**Table 1 biosensors-12-00578-t001:** The optimal concentrations of antibody and antigen for the antibody chips.

Antibody Chip	Coating Concentration (µg/mL)	Antibody Concentration (µg/mL)
Cefalexin	6	0.60
Ceftiofur	32	0.48
Neomycin	16	1.20
Gentamicin	20	0.85
Sulfonamides	10	1.40

**Table 2 biosensors-12-00578-t002:** Performance of the five mAbs and MRLs of the corresponding antibiotics.

Antibiotics	LOD (µg/L)	IC_50_ (µg/L)	Linear Ranges (µg/L)	MRLs (µg/kg)
CLX	0.84	2.54 ± 0.22	0.625–10	100
CRO	0.64	2.15 ± 0.30	0.625–10	100
N	0.51	1.97 ± 0.32	0.625–10	150
GM	4.30	15.60 ± 1.01	5–80	100
SM2	0.99	4.55 ± 0.47	1.5–24	100

**Table 3 biosensors-12-00578-t003:** The cross-reactivity rates of related analytes with the five mAbs.

Analytes	Cross-Reactivity Rates (CR, %)				
	CLX-Mab	CRO-Mab	N-Mab	GM-Mab	SM2-Mab
CLX	100.0	–	–	–	–
CFR	122.8	–	–	–	–
CE	99.1	–	–	–	–
CRO	– ^1^	100.0	–	–	–
EFT	–	109.4	–	–	–
DFC	–	92.1	–	–	–
CTX	–	9.7	–	–	–
CQO	–	5.3	–	–	–
N	–	–	100.0	–	–
AK	–	–	72.7	–	–
PRM	–	–	143.7	–	–
GM	–	–	–	100.0	–
SC	–	–	–	36.3	–
SM2	–	–	–	–	100.0
SMM	–	–	–	–	496.8
SD	–	–	–	–	525.0
SMD	–	–	–	–	641.7
SPD	–	–	–	–	64.0
SPZ	–	–	–	–	172.4
SMP	–	–	–	–	70.2
ST	–	–	–	–	156.1
SDM′	–	–	–	–	1155.0
SM1	–	–	–	–	330.0
SM2′	–	–	–	–	243.2
SQ	–	–	–	–	121.6
SDM	–	–	–	–	87.2
SMZ	–	–	–	–	18.5
SMTZ	–	–	–	–	11.8

^1^ “–” means CR is less than 0.1%.

**Table 4 biosensors-12-00578-t004:** Recoveries and LODs of antibody chip biosensors.

Drug	Samples	Recovery (%)	CV_intra-assay_ (%, *n* ^1^ = 3)	Recovery (%)	CV_inter-assay_ (%, *n* = 9)	LOD (µg/kg or µg/L)	LOQ (µg/kg or µg/L)
CLX, CFR, CE	milk	83.5–113.6	≤12.9	87.8–105.3	≤9.7	19.7–21.0	27.9–28.3
	pork	84.1–105.0	≤9.7	87.8–96.7	≤8.6	19.0–21.3	26.0–28.8
CRO, EFT, DFC	milk	81.6–113.0	≤10.4	85.9–106.0	≤9.3	20.2–21.9	28.4–31.2
	pork	83.6–110.0	≤9.2	87.9–101.2	≤9.3	17.9–20.6	25.0–28.2
N, AK, PRM	milk	84.2–110.2	≤10.3	90.3–104.7	≤8.8	13.2–23.9	18.5–30.7
	pork	84.6–107.4	≤11.9	89.2–97.5	≤9.2	12.9–23.6	17.7–29.9
GM	milk	85.9–109.3	≤10.4	93.4–100.6	≤8.5	15.4	22.7
	pork	86.7–110.2	≤9.2	92.7–102.8	≤9.1	15.1	21.5
SM2, SD, SMD, SDM′, SM1, SPD, SPZ	milk	85.8–113.1	≤10.8	90.8–105.5	≤11.6	17.4–21.0	22.1–31.1
	pork	83.7–108.2	≤10.5	88.3–105.3	≤9.9	17.5–20.4	21.8–28.4

^1^ “*n*” means the number of parallel detections.

## Data Availability

Not applicable.

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
