# Peer review of "The Simultaneous Detection of Multiple Antibiotics in Milk and Pork Based on an Antibody Chip Biosensor"

_biosensors, 2022, doi:10.3390/bios12080578_

Round 1
Reviewer 1 Report
Fluorescent immunoassay on hapten-modified chip slides for antibiotics residue in milk and meat. Already well-known approaches and based on commercially available reagents. The added value is comparison of several types of slides. Methods are well described, detailed demonstration of the performance on real samples. Not very innovative overall, though information from the routine use might be interesting for potential users.
Fig. 3A,B - join together to allow an easier comparison; the aldehyde slide, how the background can be higher than the useful signal?
Table 1, antibody dilution numbers are meaningless due to missing information on the antibody preparates used.
Reviewer 2 Report
This manuscript took detailed experiments for the detection of multiple antibiotics and compared the results with LC-MS, demonstrating that the accuracy of this detection methods. However, the advantage of this manuscript compared with related literatures is missing. Some suggestions are bellowing,
1. Since this is a biosensing related manuscript, the novelty or importance related on the sensing method or performance need be addressed clearly.
2. Please discussed why choose those parameter in detail, the selected parameters will give what effect on the performance of sensing? Why the program C gave the smallest coefficient of variation?
3. The background of figure 2 seems inconsistently.
4. It is better to provide some real date not the statistics results.
5. Please compare the sensing performance with related literatures.
Round 2
Reviewer 2 Report
This revised manuscript can be accepted now.